# Automatic Choroid Vascularity Index Calculation in Optical Coherence Tomography Images with Low-Contrast Sclerochoroidal Junction Using Deep Learning

**Roya Arian [1], Tahereh Mahmoudi [2], Hamid Riazi-Esfahani [3] , Hooshang Faghihi [3], Ahmad Mirshahi [3], Fariba Ghassemi [3], Alireza Khodabande [3], Raheleh Kafieh [1,4,\*] and Elias Khalili Pour [3]**

1    Medical Image and Signal Processing Research Center, School of Advanced Technologies in Medicine, Isfahan University of Medical Sciences, Isfahan 81746-73461, Iran
2    Department of Medical Physics and Biomedical Engineering, School of Medicine, Shiraz University of Medical Sciences, Shiraz 71348-14336, Iran
3    Retina Ward, Farabi Eye Hospital, Tehran University of Medical Sciences, Tehran 14176-13151, Iran
4    Department of Engineering, Durham University, South Road, Durham DH1 3LE, UK
\*    Correspondence: raheleh.kafieh@durham.ac.uk

**Abstract:** The choroidal vascularity index (CVI) is a new biomarker defined for retinal optical coherence tomography (OCT) images for measuring and evaluating the choroidal vascular structure. The CVI is the ratio of the choroidal luminal area (LA) to the total choroidal area (TCA). The automatic calculation of this index is important for ophthalmologists but has not yet been explored. In this study, we proposed a fully automated method based on deep learning for calculating the CVI in three main steps: 1—segmentation of the choroidal boundary, 2—detection of the choroidal luminal vessels, and 3—computation of the CVI. The proposed method was evaluated in complex situations such as the presence of diabetic retinopathy and pachychoroid spectrum. In pachychoroid spectrum, the choroid is thickened, and the boundary between the choroid and sclera (sclerochoroidal junction) is blurred, which makes the segmentation more challenging. The proposed method was designed based on the U-Net model, and a new loss function was proposed to overcome the segmentation problems. The vascular LA was then calculated using Niblack's local thresholding method, and the CVI value was finally computed. The experimental results for the segmentation stage with the best-performing model and the proposed loss function used showed Dice coefficients of 0.941 and 0.936 in diabetic retinopathy and pachychoroid spectrum patients, respectively. The unsigned boundary localization errors in the presence of diabetic retinopathy were 3 and 20.7 μm for the BM boundary and sclerochoroidal junction, respectively. Similarly, the unsigned errors in the presence of pachychoroid spectrum were 21.6 and 76.2 μm for the BM and sclerochoroidal junction, respectively. The performance of the proposed method to calculate the CVI was evaluated; the Bland–Altman plot indicated an acceptable agreement between the values allocated by experts and the proposed method in the presence of diabetic retinopathy and pachychoroid spectrum.

**Keywords:** choroidal vascularity index; optical coherence tomography images; deep learning

## 1. Introduction

The choroid is a vascularized tissue located between the retina (as the innermost layer of the eye) and the sclera (as the outermost layer of the eye). It has the highest blood flow of all tissues in the human body. This layer is responsible for supplying blood to the outer parts of the retina and the optic nerve [1]. Choroidal changes occur primarily or secondarily in many ocular diseases. The inspection of choroidal changes adjoined with information from the retina leads to better understandings of the pathogenesis of diverse diseases and how to control responses to treatments [2].

Despite the valuable clinical information that has been gained through choroidal thickness (CT), it only represents the overall choroidal structure and provides no distinctions between the stromal and luminal vascular components [2]. Recently, the choroidal vascularity index (CVI), defined as the ratio of the choroidal luminal area (LA) to the total choroidal area (TCA), has been employed to evaluate the vascular structure of choroids in different retinal and choroidal disorders [2,3]. Since CVI's introduction, many investigations have measured its effectiveness as a useful method for disease prognostication and tracking progression with positive findings [1,4–8].

Optical coherence tomography (OCT) has become a crucial tool in retinal imaging that provides noninvasive and high-resolution cross-sectional images from the retinal layers in vivo. OCT is a preferred imaging technique that is essential for the diagnosis and management of many retinal diseases associated with the choroid. Preliminary OCT imaging devices did not have adequate resolution to demonstrate the choroidal layers, particularly the sclerochoroidal junction as the outermost boundary of the choroid. The main reason was the absorption and scattering of emitted light from the OCT device by the overlying layers, especially in patients with a thick choroid (pachychoroid spectrum) [3,9]. The advent of enhanced depth imaging (EDI)-OCT and swept-source (SS)-OCT facilitated the investigation of choroidal structures such as the vascular layers and outer borders of choroid for quantitative evaluation of biomarkers such as CT and CVI [10,11]. However, even with such novel techniques, the segmentation of the choroid and sclerochoroidal junction on OCT images (B-scans) remains challenging.

The exact localization of the sclerochoroidal junction provides useful numerical information such as the CVI, which acts as a new OCT-based biomarker and can be employed for measuring and evaluating the choroidal vascular structure in chorioretinal disorders. Manual measurement of the CVI is time-consuming, prone to subjective error, and requires much effort. Thus, manual measurement procedures have been introduced [4]. Furthermore, automatic calculation of the CVI is another crucial task that includes three main steps: 1—segmentation of choroidal layer, 2—detection of choroidal luminal vessels, and 3—computation of the CVI.

Regarding the first task, previous segmentation algorithms for detecting choroidal boundaries relied on standard image-processing methods such as graph-theory-based approaches, active contour, and statistical model-based methods [12–15]. Graph-based approaches exhibited better performance than the other methods, but they suffered from a high processing time [12,14]. Recently, deep-learning-based architectures have been successfully applied in the field of biomedical image segmentation of the retina, liver, brain, pancreas, heart, and other structures [16–19].

For segmentation of the choroidal layer, patch-based approaches were presented that used CNN for the segmentation of the RPE and choroidoscleral interface (CSI) boundaries [20–22]. In other studies, a combination of deep learning and graph cut algorithms were used for choroidal boundary segmentation [23,24]. Mao et al. suggested a skip connection attention (SCA) block integrated into U-shape architectures such as U-Net and the context encoder network (CE-Net) for automated choroid layer segmentation [25]. Xu et al. applied an automated method for PED segmentation in polypoidal choroidal vasculopathy using a deep neural network [26]. Zheng et al. carried out choroid layer segmentation to obtain several evaluation parameters in swept-source OCT images from a healthy population. Residual U-Net was used to segment the choroidal boundaries [27].

Table 1 provides a comparison of the previous methods by summarizing the recent deep learning works for choroid layer segmentation and comparing them in different aspects such as the dataset, device, proposed method, loss function, metrics, and performance. This table was designed to provide an overview of the methods and was not intended to be a numerical comparison (which was not possible due to different imaging resolutions and ROIs).

**Table 1.** Summary of the previous studies in choroid layer segmentation using deep learning approaches.

| Paper | Data | Device Name | Method | Loss Function | Metrics | | Performance |
|---|---|---|---|---|---|---|---|
| Mao et al. [25] | 20 normal human subjects | Topcon DRI-OCT-1 | SCA-CENet | Not reported | Sensitivity | | 0.918 |
| | | | | | F1-score | | 0.952 |
| | | | | | Dice coefficient | | 0.951 |
| | | | | | IoU | | 0.909 |
| | | | | | Mean absolute error (MAE) (BM) (in pixels) | | 1.945 |
| | | | | | MAE (SCI) (in pixels) | | 8.946 |
| Kugelman et al. [20] | 99 children, 594 B-scans | SD-OCT | Patch-based classification (CNN-RNN) | Tverksy loss | Mean error (ME) (in pixels) | ILM | 0.01 |
| | | | | | | BM | 0.03 |
| | | | | | | CSI | −0.02 |
| | | | | | MAE (in pixels) | ILM | 0.45 |
| | | | | | | BM | 0.46 |
| | | | | | | CSI | 3.22 |
| Masood et al. [21] | 11 normal, 4 Shortsightedness, 4 glaucoma, 3 DME | Swept-source OCT | Morphological processing and CNN | Cross entropy loss | ME (in pixels) | BM | 0.43 ± 1.01 |
| | | | | | | Choroid | 2.8 ± 1.50 |
| | | | | | MAE (in pixels) | BM | 1.39 ± 0.25 |
| | | | | | | Choroid | 2.89 ± 1.05 |
| Sui et al. [23] | 912 B-scans (618 B-scans normal, 294 macular edema) | EDI-OCT | Graph-based and CNN | MSE | MAE (in pixels) | BM | 4.6 ± 4.8 |
| | | | | | | CSI | 11.4 ± 11.0 |
| Xu et al. [26] | 50 PCV patients (1800 B-scans) | SD-OCT | Dual-stage DNN | Log-loss | MAE (in μm) | BM | 5.71 ± 3.53 |
| Tsuji et al. [24] | 43 eyes from 34 healthy individual | SS-OCT | SegNet and graph cut | Not reported | Dice coefficient | | 0.909 ± 0.505 |
| He et al. [22] | 146 OCT images | SD-OCT | Patch-based CNN classifier | Focal loss | Dice coefficient | | 0.904 ± 0.055 |
| Zheng et al. [27] | 450 images from 12 healthy individual | SS-OCT | Residual U-Net | Not reported | Failure ratio less than 0.02 mm | | 68.84% |

As discussed above, the second task for automatic calculation of the CVI is the detection of choroidal luminal vessels. Histopathological assessment is the gold standard for investigating the choroidal vascular area. However, it is not a practical clinical tool because it is dependent on autopsy or biopsy samples, and due to post-fixation shrinkage and distortion, it results in the lack of repeatability and reliability. On the other hand, it is almost impossible to label the images manually due to the complex structure and the substantial amount of time required.

After comparing different image-segmentation techniques, Agrawal et al., the pioneers of CVI measurement, adopted Niblack's auto local threshold technique in many studies of the CVI while considering different healthy and pathologic conditions. Niblack's method was selected because it considers the mean and standard deviation of all the pixels in the region of interest [4]. Vupparanbina et al. [28] introduced another automated CVI analysis. Recently, Betzler et al. developed an automated platform for measuring CVI (www.cvigrid.org; accessed on 10 May 2019). The algorithm was not officially released at the time of writing this manuscript [4,29].

To the best of our knowledge, the proposed method is the first fully automated algorithm to cover the three mentioned steps of segmentation of the choroidal layer, detection of the choroidal luminal vessels, and computation of the CVI. The novelties of this work can be categorized as:

- A fully automated method with freely available code was proposed for the first time to calculate the CVI value in diabetic retinopathy and pachychoroid spectrum using deep learning methods.
- The proposed modified U-Net segmented the choroid and BM boundaries in challenging cases such as low-contrast images with thickened choroidal areas.

- The proposed loss function (weighted sum of Dice loss (DL), weighted categorical cross entropy (WCCE), and Tversky loss) was shown to overcome the imbalanced data (small foreground vs. big background).

The remainder of this paper is organized as follows: Section 2 presents the proposed method in detail, Section 3 describes the experimental results, and Section 4 discusses the results. The dataset is available at https://zenodo.org/record/7618624#.Y-K-ci_P1D8 (accessed on 13 November 2022), and the codes can be accessed at https://zenodo.org/record/7618647#.Y-K-Uy_P1D8 (accessed on 13 November 2022).

## 2. Materials and Methods

### 2.1. OCT Data and Manual Annotation

The dataset used in this study was collected at Farabi Hospital, Tehran University of Medical Sciences, Tehran, Iran. EDI-OCT images were obtained using the RTVue XR 100 Avanti instrument (Optovue, Inc., Fremont, CA, USA). Patients with diabetic retinopathy were positioned appropriately, and equally spaced high-resolution EDI-OCT B-scans at 1.5 mm ∗ 12 mm long raster patterns were captured. In patients with pachychoroid spectrum, the image-acquisition protocol was based on EDI-OCT HD line 3 mm ∗ 12 mm long B-scans. In a retrospective study, 439 raster OCT B-scans from 112 patients with diabetic retinopathy and 98 EDI-HD OCT B-scans from 44 patients with pachychoroid spectrum were selected for training and testing the model. The study was approved by the Tehran University of Medical Sciences Institutional Review Board (IRB) (IR.TUMS.FARABIH.REC.1399.017) and adhered to the tenets of the Declaration of Helsinki. Written informed consent was obtained from all participants. The subjects' demographics and clinical characteristics are reported in Table 2.

**Table 2.** Demographics and clinical characteristics of the patients.

| Data | Pachychoroid Spectrum | Diabetic Retinopathy |
|---|---|---|
| No. of patients | 44 | 112 |
| No. of B-scan Images | 98 | 439 |
| Mean age (mean ± SD) | 50.6 ± 11.2 (range: 29–74 years) | 61 ± 8 (range: 47–78 years) |
| Gender | 37 (84.1%) male | 59 (52.6%) male |
| Best corrected visual acuity (BCVA) (mean ± SD) | 0.50 ± 0.38 | 0.57 ± 0.25 |
| Subfoveal choroidal thickness (range in μm) | (265–510 μm) | (135–370 μm) |
| Resolution | 3000 × 12,000 μm | 1500 × 12,000 μm |

For manual delineation of choroidal boundaries as ground truth, raw OCT images were imported using ImageJ software (http://imagej.nih.gov/ij; accessed on 13 November 2022), which is provided in the public domain by the National Institutes of Health, Bethesda, MD, USA. Choroidal borders of all images were delineated using the polygonal selection tool in the software toolbar. The RPE–Bruch's membrane complex (BM) and the sclerochoroidal border were selected as the upper and lower margins of the choroid, respectively. The edge of the optic nerve head and the most temporal border of the image were selected as the nasal and temporal margins of the choroidal area. All manual segmentations were conducted by a skilled grader (E.K.H.) and verified by another independent grader (H.R.E.). In case of any disputes, the outlines were segmented by consensus.

### 2.2. Manual Calculation of CVI

According to the method introduced by Sonoda et al. [30], the ground truth for the CVI values were calculated using ImageJ software. For this purpose, the total choroidal area was manually selected from the optic nerve to the temporal side of the image. The selected area was added as a region of interest (ROI) with the ROI manager tool. The CVI was calculated in selected images in both the diabetic and pachychoroid spectrum groups by randomly selecting three sample choroidal vessels with lumens larger than 100 μm using

the oval selection tool in the toolbar. The average reflectivity of these areas was determined by the software. The average brightness was set as the minimum value to minimize the noise in the OCT image.

Then, the image was converted to 8 bits and adjusted with the auto local threshold of Niblack (using default parameters). The binarized image was reconverted into an RGB image, and the luminal area was determined using the color threshold tool. The light pixels were defined as the choroidal stroma or interstitial area, and the dark pixels were defined as the luminal area. The total choroidal area (TCA), luminal area (LA), and stromal area (SA) were automatically calculated [30].

Herein, we refer to the ratio of LA to TCA as the choroidal vascularity index (CVI). The CVI was calculated separately for each image.

To evaluate the inter-rater reliability of the CVI measurement, the absolute agreement model of the inter-class correlation coefficient (ICC) was employed on 20 EDI-OCT images that were initially segmented by two independent graders. A correlation value of 0.81–1.00 indicated good agreement.

Table 3 shows the acceptable inter-rater agreements in the assessment of the CVI measurement.

**Table 3.** Acceptable inter-rater agreements in assessment of CVI measurement.

|  | Interclass Correlation Coefficient (ICC) | 95% Confidence Interval (CI) |
| --- | --- | --- |
| **CVI** | 0.969 | 0.918–0.988 |

### 2.3. Fully Automated Calculation of CVI Using Deep Learning

Fully automated calculations of the CVI included three main steps: segmenting the choroidal layer, detecting the choroidal luminal vessels, and computing the CVI. These steps are elaborated in the following sections.

#### 2.3.1. Automatic Segmentation of the Choroidal Layer

In the proposed method, manual segmentation of the choroidal area was utilized to construct the ground truth masks for the choroid area. In a simple approach, two-class segmentation should be designed to divide the image into the choroid area (foreground) and the rest of the image (background), which was the most prevalent technique in similar works. However, we found that the backgrounds of the upper and lower choroid areas had different textures because they referred to different anatomical structures. Accordingly, we selected the second three-class approach to consider the intrinsic differences of the ocular structures. The second approach yielded better results after being trained with similar data, and we used it as our final segmentation approach in the next stages.

All original training images (Figure 1a) were cropped to set the most nasal side of the macula on the margins of the optic disc (Figure 1(a1)), thus eliminating the unwanted areas of the image. Two distinct models (two-class and three-class approaches) were trained separately on the cropped images.

- In the two-class segmentation approach, all pixels of the choroidal area (target class) were replaced with white pixels (Figure 1(b2)).
- In the three-class segmentation approach, the cropped image was divided into three regions (Figure 1(c2)).

The block diagram of the proposed approach is depicted in Figure 1.

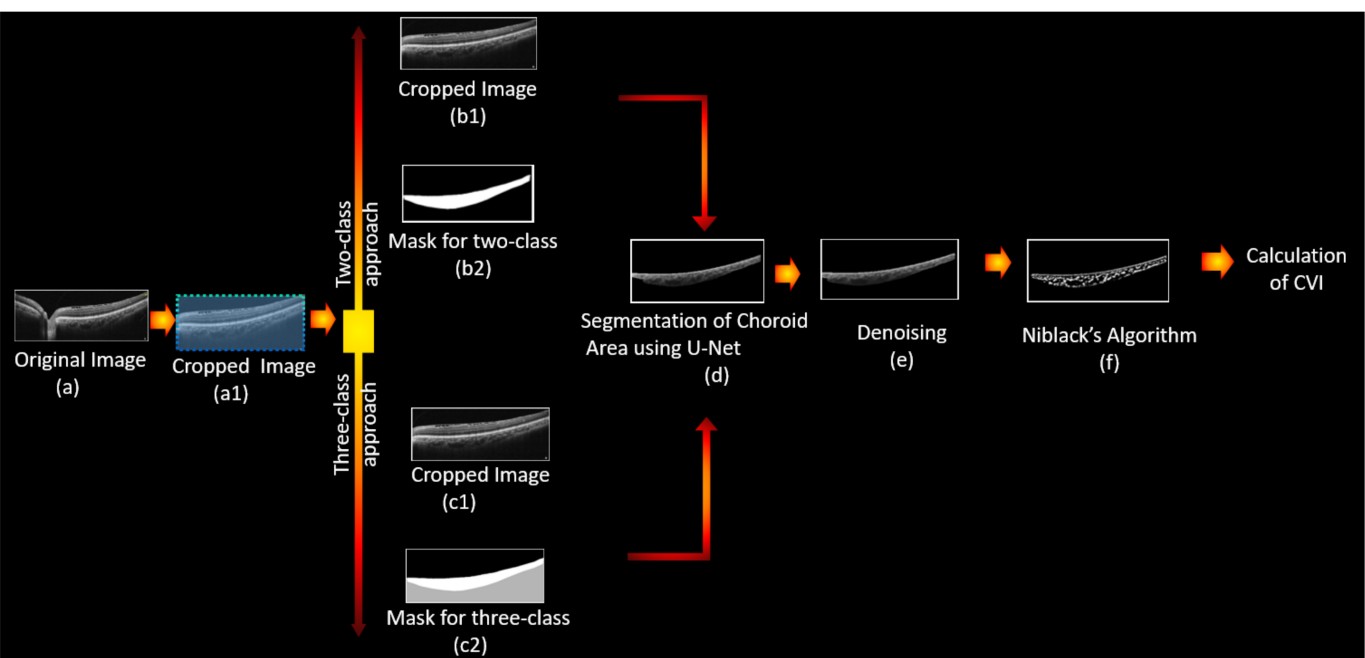

**Figure 1.** Block diagram of the proposed method. The unwanted area of the original image (**a**) is shown as cropped in (**a1**). Two approaches (two-class (**b1, 2**) and three-class (**c1, 2**)) were used for segmentation of the choroid area using U-net (**d**). After denoising of the obtained image (**e**), vascular LA detection was carried out using Niblack's auto local threshold (**f**). The CVI was finally calculated using the ratio of the LA to the TCA.

Network Architectures

A modified U-Net model with a connectivity-promoting regularization loss function was used for both the two- and three-class segmentation approaches (Figure 2). U-Net, originally proposed by Ronneberger et al., is one of the most popular segmentation models in medical imaging based on an encoder–decoder architecture and the use of skip connections [31]. The number of filters in the proposed method started at 32 and continued for five stages with a learning rate of $10^{-7}$ and an Adam optimizer. These hyperparameters were obtained empirically in order to achieve the best loss and lowest errors in the training. Due to the limited and diverse amount of data used, the high learning rate did not provide a suitable training.

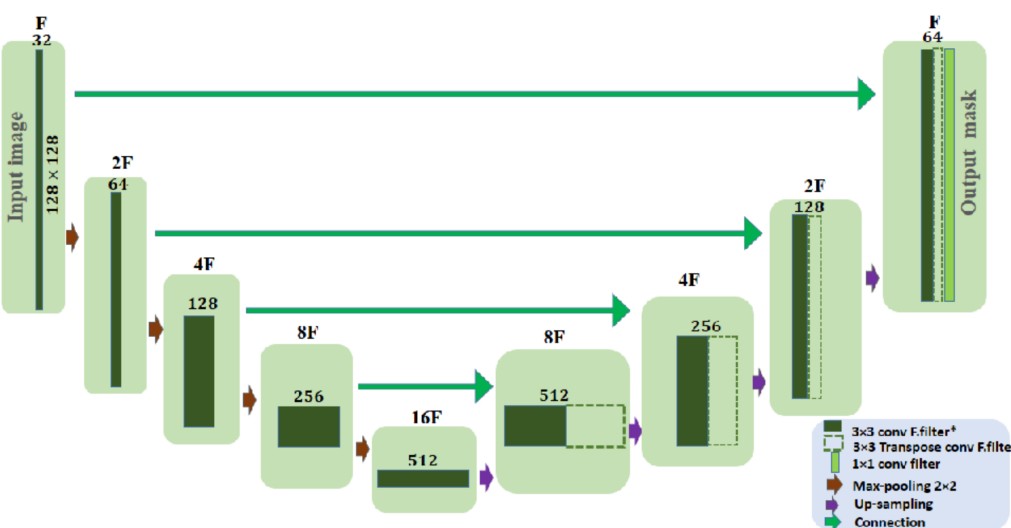

**Figure 2.** Architecture of the proposed U-Net model.

Modified Loss Function

One of the main issues in the segmentation of a relatively thin layer from a whole image is handling the unbalanced data. Standard losses such as cross-entropy loss and Dice loss usually fail on such occasions, and some new substitutes have already been introduced to solve the problem. The Dice coefficient treats false negatives (FNs) and false positives (FPs) equally, thereby resulting in higher recall and lower precision. Tversky loss [32] was introduced as an alternative to the Dice coefficient by weighing FNs more heavily than FPs for imbalanced data.

On the other hand, weighted categorical cross entropy (WCCE) was introduced to replace categorical cross entropy (CCE), the most famous loss function in classification problems. WCCE weighs the foreground more heavily than the background (choroid area vs. the whole image) and is useful for imbalanced data. To design the weights of WCCE in (j = 3)-class segmentation, we used the following equation:

$$w_j = \frac{1 - f_{w_j j}{}^\beta}{1 - \beta},\tag{1}$$

where $w_j$ is a weight vector in WCCE, $f_j$ is the ratio indicating the area corresponding to class j compared to the whole area, and $\beta$ is a hyperparameter ($\beta \in [0, 1)$) that was chosen empirically as 0.8 in this study.

Finally, considering that the desired segmented region (choroid area) had an interconnected structure and that the anatomical background did not yield disjoint sections, total variation (TV) loss was utilized to model these characteristics. To consider all of the mentioned loss functions, we investigated the results of using each individual loss function and their combinations. Four different loss functions were examined:

Loss Function #1: Dice loss
Loss Function #2: Dice loss + weighted categorical cross entropy (WCCE) [33]
Loss Function #3: Dice loss + WCCE+ total variation (TV)
Loss Function #4: Dice loss + WCCE+ Tversky

Train/Test Split and Metrics of the Segmentation Model

In health-related applications, it is crucial to split the data at the patient level (not the B-scan level) to avoid any leakage between test and training data or any correlation between the images in the test and training data. For this purpose, we divided the datasets to avoid any data from a single subject occurring in both the training and test sets. Regarding the images of diabetic retinopathy from 112 patients, 25 patients were randomly selected as the test data, and the rest were used as the training/validation data. Regarding the pachychoroid cases from 44 patients, 10 patients (20 images) were selected as test data, and the rest were employed in the training/validation data stage. Fivefold cross validation was applied to the train/validation data, and the results were reported for test sets.

For evaluating the accuracy of the segmentation, the predicted masks of the choroid layer were compared with manual masks, and the Dice similarity coefficient (DSC) was calculated based on the following equation:

$$DSC = \frac{2(A \cap B)}{A + B},\tag{2}$$

where A and B are the predicted and ground truth masks, respectively. To report an accurate Dice coefficient concentrated on the area of interest and while considering that the choroid layer was significantly smaller than the background, the DSC value was only reported for the segmented choroid layer rather than the background. The average DSC on all B-scans of the test dataset is reported in Section 3. Furthermore, the mean absolute error between the predicted boundaries and the manually annotated boundaries were calculated and reported.

2.3.2. Detection of Choroidal Luminal Vessels and Computation of the CVI

After calculation of the BM boundary and sclerochoroidal junction based on the choroidal layer mask from the U-Net architecture, the next steps were designed:

- Noise reduction using a non-local means algorithm with a deciding filter strength of 10 [34]. As shown in Figure 3, noise reduction reduced the errors in finding vascular components by omitting and noisy and disturbing pixels. Moreover, as was mentioned in Section 2.2, in order to calculate the CVI ground truth after finding the choroidal area, noise reduction was performed manually (before applying the Niblack algorithm). Therefore, in the proposed automatic method, we attempted to mimic what is already done in the manual process and achieve the most reliable performance.
- Vascular LA detection using Niblack's auto local threshold method using Python software. The selected parameters for Niblack's method are summarized in Table 4 to make the provided code reproducible. The parameters were empirically adjusted to resemble the gold standard values as much as possible in the training dataset.
- Calculation of CVI using:

$$CVI = \frac{\text{vascular luminal area}}{\text{total choroidal area}}, \tag{3}$$

Note that the total choroidal area was obtained by computing the sum of all the pixels between the two boundaries, while the vascular luminal area was the sum of all pixels that were obtained using Niblack's method.

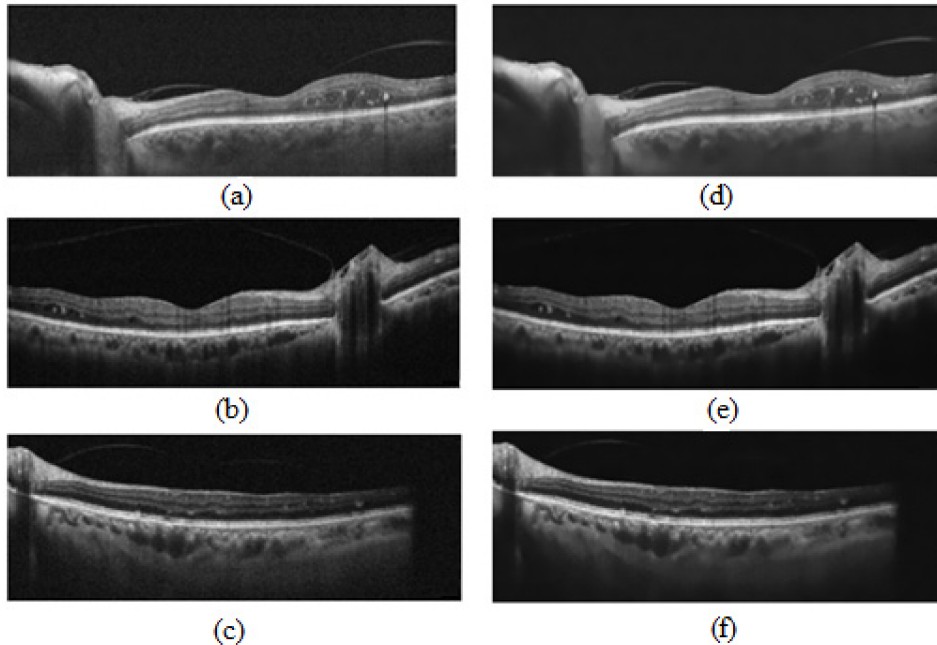

**Figure 3.** Comparing noisy and denoised images in a fully automatic method: (**a–c**) show noisy images while (**d–f**) are their respective denoised versions.

**Table 4.** Parameters used for Niblack's algorithm in two datasets.

| Parameter of Niblack's Algorithm | Diabetic Retinopathy Images | Pachychoroid Spectrum Images |
| --- | --- | --- |
| Window Size | 15 | 13 |
| K | 0.001 | 0.01 |
| Threshold Coefficient | 1.03 | 1.02 |

## 3. Results

The software environment used in this study consisted of the Keras platform backend in Python 3.7 and an NVIDIA GeForce GTX 1080Ti GPU. The performances of different proposed segmentation methods were compared while considering the values of the signed error (S_E) and unsigned error (U_E) for localization of the BM boundary and sclerochoroidal junction. These errors were calculated separately for each boundary by subtracting the pixel number of the predicted and ground truth layer in the vertical direction; these were ultimately converted to micrometers. Furthermore, the DSC for each method was reported and compared. In addition, the calculation of the CVI values was evaluated based on the Bland–Altman plot of the manual and automated measurements.

### 3.1. Choroidal Boundary Segmentation in Pachychoroid Spectrum Dataset

We compared the performances of different loss functions for two-class and three-class segmentation approaches for localization of the choroidal boundary in pachychoroid spectrum patients. The segmentation results for each loss function (using two examples) are demonstrated in Figure 4. Manually annotated boundaries for each image are shown in red, and the predicted ones are shown in blue.

For further validation purposes, the mean signed and unsigned boundary positioning errors for each layer are reported in Table 5. The errors are reported in micrometers (while considering that the height of B-scans for pachychoroid spectrum dataset was 3 mm) for input images resized to $128 \times 128$ pixels and using a batch size of 8. The experimental results with different batch sizes and different input image sizes ($64 \times 64$ and $265 \times 256$) are reported in the Supplementary Materials, and the best values for the batch size and input image size were selected for the rest of the analysis.

Note that because U-Net inherently performs much better for square images than rectangular ones, we resized the input images to different square sizes such as $64 * 64$, $128 * 128$, or $256 * 256$ pixels. Obviously, the higher the resolution that was chosen, the better the performance of model (with more time complexity). So, we used $128 * 128$ pixels as the starting resolution to save time in our long study process of investigating different hyperparameters and loss functions to find out the best ones; however, finally we tried the best-found model with different input image sizes. The results are reported in the Supplementary Materials.

**Table 5.** Mean signed and unsigned errors (S_E and U_E) for different loss functions in two- and three-class segmentation of the pachychoroid spectrum dataset (errors are reported in micrometers; the height of the B-scans for the pachychoroid spectrum dataset was 3000 µm). Bold font indicates the best results.

| Unit | Loss Function | Two-Class Segmentation | | | | Three-Class Segmentation | | | |
| --- | --- | --- | --- | --- | --- | --- | --- | --- | --- |
| | | BM Boundary | | Choroid Boundary | | BM Boundary | | Choroid Boundary | |
| | | U_E | S_E | U_E | S_E | U_E | S_E | U_E | S_E |
| µm | Loss#1 | 44.7 | −9.6 | 95.7 | −37.8 | 22.2 | −16.5 | 91.5 | 16.8 |
| | Loss#2 | 33.9 | −31.8 | 92.7 | 22.5 | 21.9 | −19.8 | 78 | 12.6 |
| | Loss#3 | 39 | 3.9 | 87.3 | 11.7 | 22.5 | −11.1 | 78.3 | 15.3 |
| | Loss#4 | 30.3 | −25.5 | 84.3 | −5.7 | **21.6** | −14.4 | **76.2** | −25.5 |

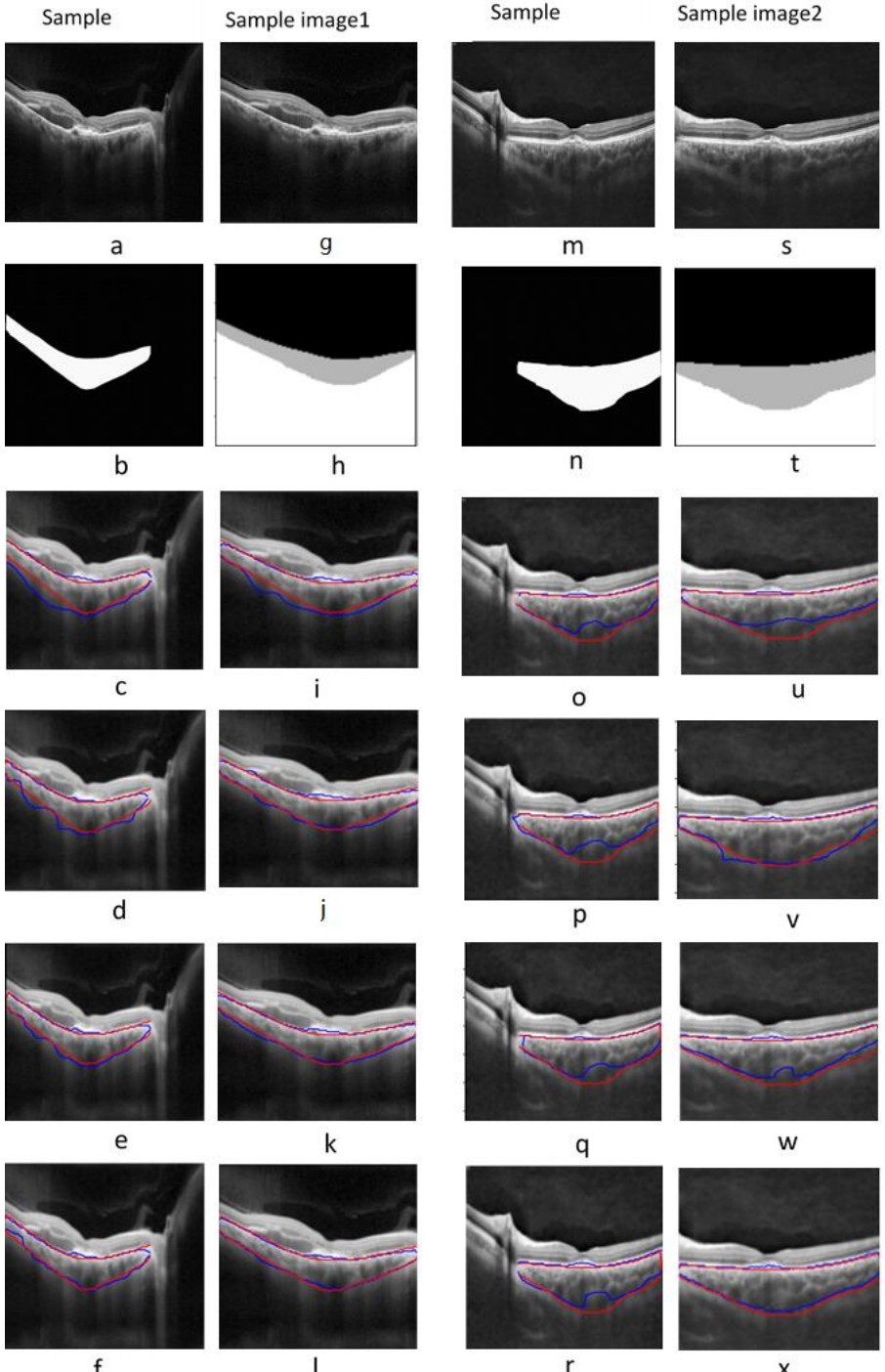

**Figure 4.** Segmentation results of the two example images in the pachychoroid spectrum class. (**a**) Original pachychoroid spectrum sample 1; (**b**) two-class ground truth. Real and predicted boundaries of two-class approach using loss function: (**c**) DL; (**d**) DL + WCCE; (**e**) DL + WCCE + TV; (**f**) DL + WCCE + Tversky. (**g**) Cropped original pachychoroid spectrum sample 1; (**h**) three-class ground truth. Real and predicted boundaries of three-class approach using loss function: (**i**) DL; (**j**) DL + WCCE; (**k**) DL + WCCE + TV; (**l**) DL + WCCE + Tversky. (**m**) Original pachychoroid spectrum sample 2; (**n**) two-class ground truth. Real and predicted boundaries of two-class approach using loss function: (**o**) DL; (**p**) DL + WCCE; (**q**) DL + WCCE + TV; (**r**) DL + WCCE + Tversky. (**s**) Cropped original pachychoroid spectrum sample 2; (**t**) three-class ground truth. Real and predicted boundaries of three-class approach using loss function: (**u**) DL; (**v**) DL + WCCE; (**w**) DL + WCCE + TV; (**x**) DL +WCCE + Tversky. Manually annotated boundaries for each image are shown in red and predicted ones are shown in blue.

### 3.2. Choroidal Boundary Segmentation in the Diabetic Retinopathy Dataset

Similar to the pachychoroid spectrum dataset, the segmentation results for the diabetic retinopathy dataset are shown in Figure 5, and the boundary positioning errors are reported in Table 6. The errors are reported in micrometers (while considering that the height of B-scans for diabetic retinopathy dataset was 1.5 mm) for input images resized to 128 × 128 pixels. The experimental results with different batch sizes and different input image sizes (64 × 64 and 265 × 256) are reported in the Supplementary Materials, and the best values for the batch size and input image size were selected for the rest of the analysis.

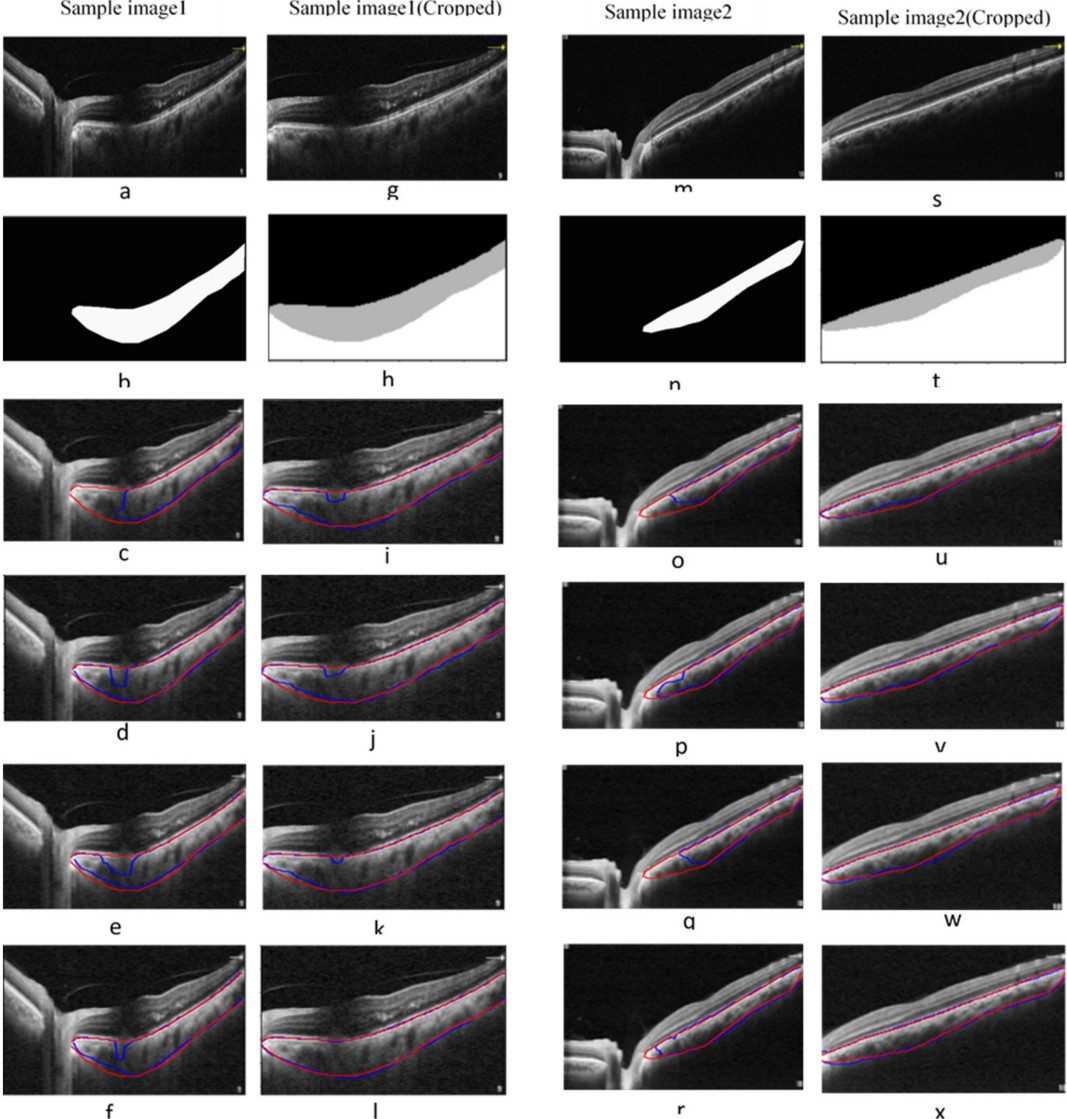

**Figure 5.** (**a**) Original diabetic retinopathy subject 1; (**b**) two-class ground truth. Real and predicted boundaries of two-class approach using loss function: (**c**) DL; (**d**) DL + WCCE; (**e**) DL + WCCE + TV; (**f**) DL + WCCE + Tversky. (**g**) Cropped original diabetic retinopathy subject 1; (**h**) three-class ground truth. Real and predicted boundaries of three-class approach using loss function: (**i**) DL; (**j**) DL + WCCE; (**k**) DL + WCCE + TV; (**l**) DL + WCCE + Tversky. (**m**) Original diabetic retinopathy subject 2; (**n**) two-class ground truth. Real and predicted boundaries of two-class approach using loss function: (**o**) DL; (**p**) DL + WCCE; (**q**) DL + WCCE + TV; (**r**) DL + WCCE + Tversky. (**s**) Cropped original diabetic retinopathy subject 2; (**t**) three-class ground truth. Real and predicted boundaries of three-class approach using loss function: (**u**) DL; (**v**) DL + WCCE; (**w**) DL + WCCE + TV; (**x**) DL + WCCE + Tversky. Manually annotated boundaries for each image are shown in red and predicted ones are shown in blue.

**Table 6.** Mean signed and unsigned errors (S_E and U_E) for different loss functions in two- and three-class segmentation of the diabetic retinopathy dataset (errors are reported in micrometers; the height of the B-scans for the diabetic retinopathy spectrum dataset was 1500 μm). Bold font indicates the best results.

| Unit | Loss Function | Two-Class Segmentation | | | | Three-Class Segmentation | | | |
| | | BM Boundary | | Sclerochoroidal Boundary | | BM Boundary | | Sclerochoroidal Boundary | |
| | | U_E | S_E | U_E | S_E | U_E | S_E | U_E | S_E |
|---|---|---|---|---|---|---|---|---|---|
| μm | Loss#1 | 30.15 | 6.75 | 45.15 | 19.35 | 4.95 | −4.35 | 21.45 | −3.3 |
| | Loss#2 | 22.5 | 3.45 | 37.2 | 14.55 | 3.15 | −1.5 | 20.85 | 0.45 |
| | Loss#3 | 22.8 | 4.8 | 37.65 | 16.5 | 3.15 | −1.05 | 22.2 | −7.35 |
| | Loss#4 | 21.15 | 1.65 | 31.35 | −0.9 | **3** | 0.15 | **20.7** | 0.15 |

In addition, the DSC value between the segmented choroid layer and the ground truth is shown in Figure 6 for each combination of loss functions. The results indicated that the best performance was exhibited by the three-class segmentation and a combination of the best-chosen loss function (DL, WCCE, and Tversky losses). The sample results using the best-selected model are demonstrated in Figure 7 for diabetic retinopathy and pachychoroid spectrum data.

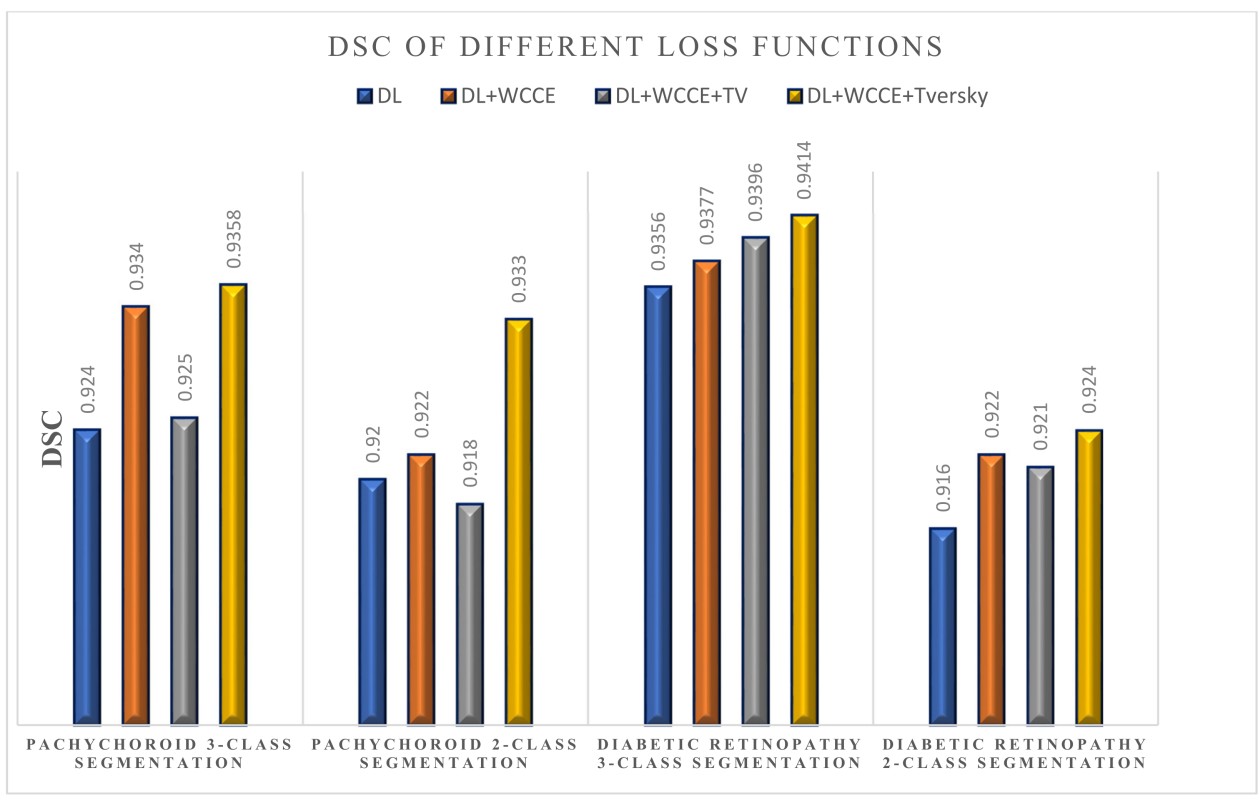

**Figure 6.** Comparison of DSC value between the segmented choroid layer and the ground truth for different loss functions in the pachychoroid spectrum and diabetic retinopathy datasets using two- and three-class segmentation.

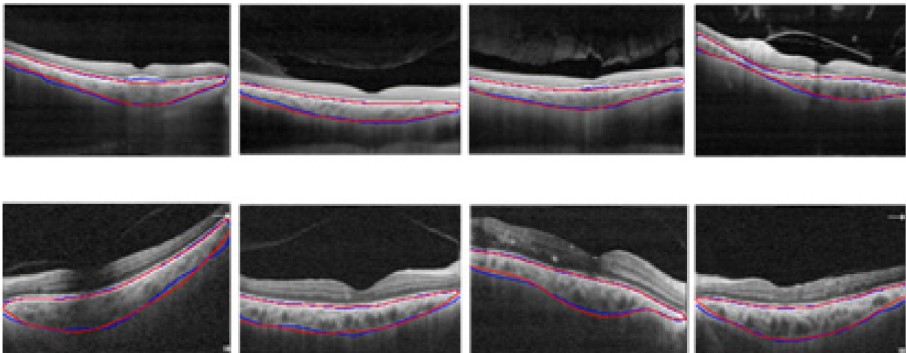

**Figure 7.** Illustrations of the four samples with final segmentation using the three-class approach with a combination of three loss functions (DL + WCCE + Tversky). The first row belongs to the diabetic retinopathy dataset, and the second row shows the results from the pachychoroid spectrum dataset. Manually annotated boundaries for each image are shown in red and predicted ones are shown in blue.

As mentioned previously, deep-learning-based architectures have been successfully applied recently in the field of biomedical image segmentation [16–19]. Therefore, in order to compare our proposed method with other segmentation methods, we employed only deep-learning-based methods that included semantic segmentation and the path-based methods mentioned in [20]. According to the limited data used in this study, most of the complex deep learning methods cannot be trained by such a small amount of training data, and in the best case the Dice coefficient will be equal to 60%. To overcome this problem, first we trained the models using the main data of the study and then evaluated them using our test dataset. The results are compared in Table 7.

**Table 7.** Comparison of different deep learning segmentation algorithms. Bold font indicates the best results.

| Unit | Method | Pachychoroid | | | | Diabetic Retinopathy | | | |
| | | BM Boundary | | Choroid Boundary | | BM Boundary | | Choroid Boundary | |
| | | U_E | S_E | U_E | S_E | U_E | S_E | U_E | S_E |
|---|---|---|---|---|---|---|---|---|---|
| μm | Proposed model | **21.6** | −14.4 | **76.2** | −25.5 | **3** | 0.15 | **20.7** | 0.15 |
| | Semantic segmentation [20] | 27.1 | −5.76 | 88.5 | 45.7 | 4.5 | 1.56 | 29.3 | −4.67 |
| | Patch-based [20] | 32.2 | 25.3 | 96.5 | 4.32 | 6.1 | −3.4 | 40.1 | 0.26 |

### 3.3. Vascular LA Segmentation CVI Measurement

The results of the vascular LA segmentation method are shown in Figure 8. The automated and manual CVI for pachychoroid spectrum and diabetic retinopathy were measured, and the Bland–Altman limits of agreement between the manual and automated measurements were calculated for the pachychoroid spectrum and diabetic retinopathy data. Bland–Altman plots were used to graphically represent the agreement between the manual and automated methods in the CVI measurements as shown in Figure 9. The limits of agreement between the automated and manual methods to measure the CVI ranged from −0.168 to 0.120 and −0.080 to 0.095 for pachychoroid spectrum and diabetic retinopathy data, respectively.

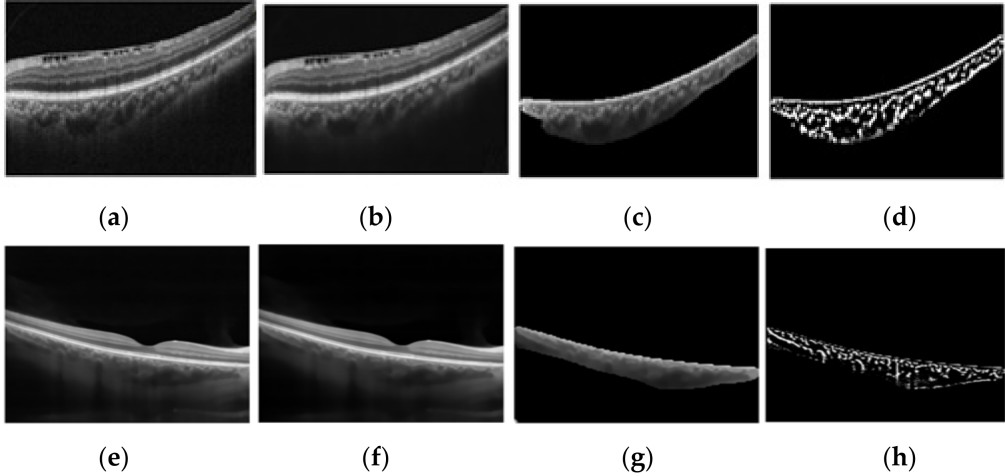

**Figure 8.** Illustration of the vascular LA detection using Niblack's algorithm. First row: (**a**) original image (diabetic retinopathy); (**b**) denoised image; (**c**) predicted choroid area; (**d**) vascular LA detection. Second row: (**e**) original image (pachychoroid spectrum); (**f**) denoised image; (**g**) predicted choroid area; (**h**) vascular LA detection.

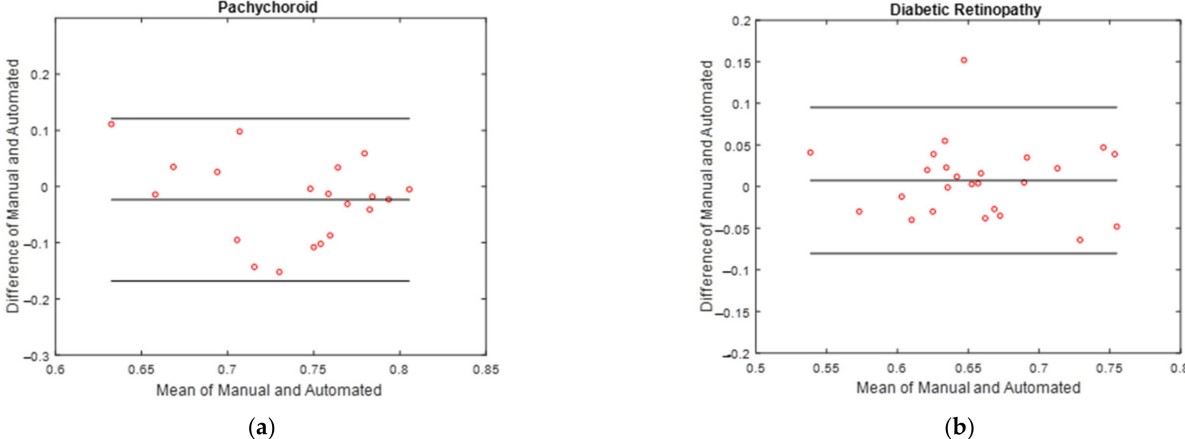

**Figure 9.** Bland–Altman plots of manual and automated measurements of CVI in (**a**) pachychoroid spectrum and (**b**) diabetic retinopathy data.

## 4. Discussion

The recently introduced CVI metric was developed in response to the need for choroidal vasculature assessments that are more reliable and precise than previous measures such as CT and choroidal vessel diameter, which have limitations. CVI displayed less changeability than CT and was influenced by fewer physiologic elements in previous studies, which showed that it is a generally stable biomarker for studying choroidal modifications [4–6].

The need for automatic calculations of the CVI is undeniable in the presence of a high number of OCT B-scans; in addition, there is no fully automatic method with freely available code to address this issue. In the current study, a U-Net architecture was used to segment the RPE boundary and sclerochoroidal junction. A tailored set of loss functions with a combination of four classic losses (such as the Dice loss coefficient (DL), weighted categorical cross-entropy (WCCE), total variation (TV), and Tversky) were examined to fulfill the need to manage the imbalanced data in the segmentation of small regions in an image.

Subsequently, the desired task was inspected using two-class and three-class segmentation approaches. At first, we carried out a two-class approach. One class was the choroid area, and the rest of the image was the other class. As the upper and lower choroid

areas had different textures, we used a three-class approach as the final approach. The experimental results showed that the three-class approach performed better at segmenting the choroid area, so we used this network for the final segmentation to obtain the CVI. The experimental results with three-class segmentation using combined loss (Dice, WCCE, and Tversky) showed Dice coefficients of 0.941 and 0.936 in the diabetic retinopathy and pachychoroid spectrum patients, respectively (Figure 5).

In the presence of diabetic retinopathy, the unsigned boundary localization errors were 3 and 20.7 μm for the BM boundary and sclerochoroidal junction, respectively. However, in the presence of the pachychoroid spectrum, the unsigned errors were 21.6 and 76.2 μm for BM and sclerochoroidal junction, respectively (Tables 4 and 5).

Then, Niblack's auto local threshold was adopted to calculate the vascular LA in accordance with clinically approved publications. The CVI value was then calculated based on this stage, and the Bland–Altman plots indicated an acceptable agreement between the manually allocated values and the proposed method for CVI calculation.

The results of our study showed that the designed deep learning algorithm could detect the RPE with a higher accuracy than the sclerochoroidal junction due to the higher intensity of the RPE signal in the OCT images compared to the sclerochoroidal junction.

Previous studies were limited to normal OCT B-scans, and the challenge of segmenting blurred boundaries was achieved in this work. There is just one available code called OCT-tool [35] that can be used to segment the choroidal layer. Using this code, the BM boundary of each test datum was automatically detected; however, the choroid boundary was detected manually by adding some control points. It took a long time to add enough control points to fit the choroid layer. However, in the current study this layer was automatically detected with a high accuracy, which was greatly time consuming.

Due to the uncertainty of the sclerochoroidal junction, choroidal segmentation is difficult in patients with a thick choroidal (such as patients with pachychoroid spectrum). Therefore, in this study, in addition to diabetic patients, a group of patients with pachychoroid spectrum was included. Moreover, we considered ROI extended from the optic nerve to the temporal side of the image to obtain a more precise estimate of the CVI. While Agrawal et. al. [6] demonstrated that the subfoveal CVI could accurately reflect the CVI of the entire macular area in healthy individuals, this assertion did not hold in pachychoroid disease or ethnicities with a thicker choroid.

The limitations of our study included the following. This study was performed on just a small number of OCT B-scans and included only two disease entities: diabetic retinopathy and pachychoroid spectrum. At best, the accuracy of our algorithm was in the range of 10 microns around the sclerochoroidal junction in both groups of patients with diabetic retinopathy and pachychoroid spectrum. Although this accuracy seemed acceptable for patients in the current study, it may not be satisfactory for patients with a very thin choroid (such as patients with age-related macular degeneration). The same is true for patients with very thick choroidal structures (such as patients with an infiltrative choroidal disease; e.g., Vogt-Koyanagi-Harada). Furthermore, blood vessel reflection and shadowing is a potential source of errors in cross-sectional OCT images for choroidal boundary segmentation, thus limiting the accuracy of both automated and manual approaches for precise segmentation and CVI measurement.

Moreover, the OCT B-scans used in the current study were captured with an RTVue XR 100 Avanti instrument. However, the results of a previous study by Agrawal et al. showed that CVI measurement was not affected by the device used (SD-OCT vs. swept-source) [36]. However, to generalize the results of deep learning algorithms to other diseases, these algorithms should be evaluated using diverse devices in different diseases.

Similar to previous studies in this area, the current study assessed the choroid in only a limited cross-sectional B-scan that passed through the macular area, and it is critical not to extend the current findings to all regions of the retina and choroid (especially peripheral sections). Future advancements in OCT devices capable of ultra-wide field imaging as well as 3D assessments of these regions will improve the accuracy of assessments.

In conclusion, our limited experience showed that a deep-learning-based algorithm for the automatic calculation of the CVI in OCT images of patients suffering from diabetic retinopathy and pachychoroid spectrum can yield a good performance in this regard. Larger-scale studies on different retinochoroidal diseases with more OCT B-scans captured with different devices can improve automated measurements of the CVI, which is a useful biomarker for diagnosis and prognostication.

**Supplementary Materials:** The following supporting information can be downloaded at: https://www.mdpi.com/article/10.3390/photonics10030234/s1.

**Author Contributions:** Conceptualization, T.M., H.R.-E. and E.K.P.; Methodology, T.M., R.K. and E.K.P.; Software, R.A.; Validation, H.R.-E. and E.K.P.; Formal analysis, R.A.; Investigation, R.A. and T.M.; Resources, H.R.-E. and E.K.P.; Data curation, H.R.-E., H.F., A.M., F.G., A.K. and E.K.P.; Writing—original draft, R.A. and T.M.; Writing—review & editing, H.R.-E., H.F., A.M., F.G., A.K., R.K. and E.K.P.; Supervision, R.K.; Project administration, H.R.-E. and E.K.P. All authors have read and agreed to the published version of the manuscript.

**Funding:** This work was supported in part by the Vice-Chancellery for Research and Technology of Isfahan University of Medical Sciences under Grant 240046 and by the National Institute for Medical Research Development under Grant 964582.

**Institutional Review Board Statement:** The study was approved by the Tehran University of Medical Sciences Institutional Review Board (IRB) (IR.TUMS.FARABIH.REC.1399.017) and adhered to the tenets of the Declaration of Helsinki. Written informed consent was obtained from all participants.

**Data Availability Statement:** The dataset is available at https://zenodo.org/record/7618624#.Y-K-ci_P1D8 (accessed on 7 February 2023), and the codes can be accessed at https://zenodo.org/record/7618647#.Y-K-Uy_P1D8 (accessed on 7 February 2023).

**Conflicts of Interest:** The authors declare no conflict of interest.

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
