# Peer review of "Automatic Choroid Vascularity Index Calculation in Optical Coherence Tomography Images with Low-Contrast Sclerochoroidal Junction Using Deep Learning"

_photonics, doi:10.3390/photonics10030234_

Round 1
Reviewer 1 Report
The manuscript submitted as “Automatic Choroid Vascularity Index Calculation in Optical Coherence Tomography Images with Low Contrast Sclerochoroidal Junction Using Deep Learning” proposed a method to calculate choroid vascularity index (CVI) in OCT images with segmentation of the choroidal boundary using modified UNet and the detection of the choroidal luminal vessels. The paper is well organized with adequate information and the experimental results are quite promising. However, some minor revisions of the paper are required for publication. Detailed suggestions and questions are listed below:
1. In line 72, “manual measurement of CVI is prone to objective error” is opposite to the empirical knowledge of subjective biases from manual measurements. You may explain more here.
2. In the pipeline of the proposed method, you denoised the image before detecting the choroidal luminal vessels without giving more information about how the denoising part is carried out. It is suggested to illustrate a bit about the denoising method you used and maybe even more about how it affects the results.
3. In line 221, you chose a rather small learning rate as for Adam optimizer. With such a small learning rate, the training process usually converges rather slowly. Is the learning rate consistent in the whole training stage or there’s a decay? Did the slow convergence occur to you and how did you handle it?
4. In line 319, you mentioned the input images are resized to 128128 pixels for segmentation. I would assume that the choroidal luminal vessels detection and CVI calculation after are based on the image size of 128128 pixels as well. You also mentioned the original EDI-OCT images with high resolution is critical for choroid layers detection in line 60. How do you think it would affect the final results when you process the image with such low resolution?
Author Response
Detailed answer to the comments is attached.

Reviewer 2 Report
The authors performed the automatic calculation of CVI in OCT images by a deep learning-based algorithm. They used two-class and three-class approaches to extract the choroid area. Then they conducted four loss functions or a combination to achieve the minimum error for the datasets. The methods and their result are interesting and significant. I recommend the manuscript to be published with minor revision.
1. What is the accuracy of manually annotated boundaries? Is there any error in determining these boundaries?
2. How are the boundary positioning errors determined? Some of the real and predicted boundaries are not closed loops. Do they influence the positioning errors or the choroid area?
3. Page 13 line 404: ......to measure CVI ranged from -0.168 to 0.120 to and -0.080 404 to 0.095 to for pachychoroid spectrum... The sentence needs to be rephrased.
Author Response

(The authors gave the same response as above.)

Reviewer 3 Report
In this paper, the authors present a method for Choroid Vascularity Index calculation based on deep learning. I have the following concerns about this paper. 1) It is not clear how do the authors obtain the ground truth for choroidal luminal vessels. 2) There are many existing methods for choroidal layer segmentation, the authors should compare with them to verify the effectiveness of the proposed method in choroidal layer segmentation. 3) The data seems not publicly available at present.
Author Response

(The authors gave the same response as above.)
